# UAVSwarm Dataset: An Unmanned Aerial Vehicle Swarm Dataset for Multiple Object Tracking

Chuanyun Wang [1], Yang Su [2], Jingjing Wang [3], Tian Wang [4] and Qian Gao [1,*]

1   College of Artificial Intelligence, Shenyang Aerospace University, Shenyang 110136, China; wangcy0301@sau.edu.cn
2   School of Computer Science, Shenyang Aerospace University, Shenyang 110136, China; suyang1@stu.sau.edu.cn
3   China Academic of Electronics and Information Technology, Beijing 100041, China; wangjingjing@cetccloud.com
4   Institute of Artificial Intelligence, Beihang University, Beijing 100191, China; wangtian@buaa.edu.cn
*   Correspondence: gaoqian@buaa.edu.cn

**Abstract:** In recent years, with the rapid development of unmanned aerial vehicles (UAV) technology and swarm intelligence technology, hundreds of small-scale and low-cost UAV constitute swarms carry out complex combat tasks in the form of ad hoc networks, which brings great threats and challenges to low-altitude airspace defense. Security requirements for low-altitude airspace defense, using visual detection technology to detect and track incoming UAV swarms, is the premise of anti-UAV strategy. Therefore, this study first collected many UAV swarm videos and manually annotated a dataset named UAVSwarm dataset for UAV swarm detection and tracking; thirteen different scenes and more than nineteen types of UAV were recorded, including 12,598 annotated images—the number of UAV in each sequence is 3 to 23. Then, two advanced depth detection models are used as strong benchmarks, namely Faster R-CNN and YOLOX. Finally, two state-of-the-art multi-object tracking (MOT) models, GNMOT and ByteTrack, are used to conduct comprehensive tests and performance verification on the dataset and evaluation metrics. The experimental results show that the dataset has good availability, consistency, and universality. The UAVSwarm dataset can be widely used in training and testing of various UAV detection tasks and UAV swarm MOT tasks.

**Keywords:** unmanned aerial vehicles (UAV) swarm; multiple object tracking; unmanned aerial vehicles (UAV) detection; image dataset

## 1. Introduction

With the accelerated transformation of modern warfare forms to information, unmanned, and intelligent, unmanned aerial vehicles (UAV) have been widely used in intelligence, reconnaissance, surveillance, interference, decoy, precision strike, damage assessment and other operational tasks in the military field [1]. Due to the complex and changeable battlefield environment, the combat task of strong confrontation poses new challenges to the wartime survivability and task execution ability of a single UAV [2]. Therefore, the combat style of conducting complex combat missions in the form of ad hoc networks, consisting of hundreds of small-size and low-cost UAV, is attracting great attention and extensive research [3]. UAV swarm warfare uses multiple UAV to cooperate, through complementary capabilities and action coordination, break through the enemy tight air defense circle, complete complex intelligence, surveillance, and reconnaissance tasks, as well as collaborative attack and damage assessment tasks, showing a high level of coordination and intelligence [4].

High-altitude high-speed, stealth long flight, micro cluster, intelligent new concept UAV swarms will play an increasingly important role. Therefore, the problem of defense UAV swarms in low altitude airspace is attracting great attention, and the technology of

detecting and early warning attacking the UAV swarm is being widely studied. Due to the characteristics of there being a large number, distributed, no center, self-organization, low cost, flexible, cooperative penetration, and cooperative attack, the low-altitude airspace defense UAV swarm is facing great challenges. At present, the low-altitude airspace defense UAV swarm methods mainly include interference influence, violent destruction, and reconnaissance control. However, the premise of the above anti-UAV strategy is to detect, identify, track, and locate the invading UAV swarm effectively. Radar detectors have been widely used in UAV detection and tracking due to their long detection distance, high sensitivity, and all-weather adaptability [5–8]. However, the high price, poor flexibility, and poor concealment of radar detectors seriously restricts their application scenarios and combat effectiveness. In recent years, low-cost, high-resolution visual sensor technology has developed rapidly. Infrared, visible light and other visual detection technologies have shown excellent performance in UAV detection and tracking, which has attracted more and more attention from researchers [9–11]. The main advantages of visual detection technology include intuitive detection results, low system cost, fast detection speed, long detection distance, and high applicable scenes. These advantages determine that visual detection technology is an integral part of the low-altitude airspace defense UAV swarm.

Using visual technology to detect and track UAV swarm is to locate and classify UAV objects in infrared or visible images and videos, and then implement multi-object tracking (MOT). Although there are many different methods to solve the problem of MOT, due to the particularity of UAV swarm flying in three-dimensional space, it brings challenges such as sudden appearance changes, serious object occlusion, and frequent field of view. In recent years, with the rapid development and extensive research of deep learning technology, the accuracy of the object detection algorithm is continuously improved [12,13], and the MOT algorithm based on detection has also been greatly developed [14]. Detection-based MOT can be divided into online tracking and offline tracking. The representative online MOT algorithms are SORT [15], DeepSORT [16], MOTDT [17], JDE [18] and Fair [19]. Representative off-line MOT algorithms include POI [20], IOU [21], LMP [22], and multi-cue-based MOT [23], etc.

The advanced MOT models are mainly data driven, which depend on large-scale databases. The well-labeled datasets have proved to be of profound value for the effectiveness and accuracy in various MOT tasks. The open dataset and benchmark applicable to UAV swarm MOT tasks have not been reported, which limits the development of anti-UAV strategy using visual detection technology. Thus, the first step of detecting and tracking UAV swarm is to build up a dataset of UAV swarm. In this study, an image dataset named the UAVSwarm dataset is constructed, which records 13 different scenarios and more than 19 types of UAV, including 12,598 manually annotated images; the number of UAV in each sequence is 3 to 23, and the dataset can be widely used in UAV swarm detection and tracking tasks. To maintain the universality and robustness of the trained models, two advanced depth detection models are used as strong benchmarks, namely Faster R-CNN and YOLOX. Then, two state-of-the-art MOT models, GNMOT and ByteTrack, are used to conduct comprehensive tests and performance verification on the dataset and evaluation metrics.

The remainder of this paper is organized as follows. Firstly, we review the related works in Section 2. Then, Section 3 presents the proposed UAVSwarm dataset. In Section 4, several experiments are carried out to demonstrate the effectiveness of the proposed dataset and explain its performance with baseline metrics. Finally, conclusions are drawn in Section 5.

## 2. Related works

### 2.1. Image-Based UAV Datasets

#### 2.1.1. Real Word Dataset

Pawełczyk et al. [24] expanded existing multiclass image classification and object detection datasets (ImageNet, MS-COCO, PASCAL VOC, anti-UAV) with a diversified

dataset of drone images. To maximize the effectiveness of the model, real world footage was utilized, transformed into images and hand-labelled to create a custom set of 56,821 images and 55,539 bounding boxes.

Compared with other UAV datasets, the Real Word dataset contains the most types of UAV and environments, and the image resolution is low, because all the data are obtained from YouTube videos, while other datasets are collected by researchers themselves. Due to the limitation of shooting perspective, most of the data in the Real World dataset is in flat view and elevation.

### 2.1.2. Det-Fly Dataset

Zheng et al. [25] present a new dataset, named Det-Fly, which consists of more than 13,000 images of a flying target UAV acquired by another flying UAV. Compared to the existing datasets, the Det-Fly dataset is more comprehensive in the sense that it covers a wide range of practical scenarios with different background scenes, viewing angles, relative distance, flying altitude, and lightning conditions.

The Det-Fly dataset overcomes the shortcomings of UAV data from a single perspective. The camera collects the target UAV directly in the air, including a variety of UAV postures under elevation, pitch, and horizon. However, the data set contains only one type of UAV, so that the model cannot be used for other types of UAV detection.

Regarding the other image-based UAV dataset, the size of UAV in the MIDGARD dataset [26] is relatively large, and the appearance and outline of UAV are very clear. The camera and UAV are very close, but it is impossible to study the detection problem in long distance. The images in the USC-Drone dataset [27] are taken by people standing on the ground with hand-held cameras, and the UAV's perspective is too single. In addition, there are many unmarked images in this dataset that cannot be used directly. These datasets also contain only one type of UAV and a richer environment, but also has the disadvantage of a single viewing angle, so cannot be used for MOT tasks.

### 2.2. Video-Based UAV Datasets

### 2.2.1. Purdue Dataset

The study by Jing et al. [28] comprises five video sequences of 1829 frames with 30 fps frame rate. The Purdue dataset is recorded by a GoPro 3 camera (HD resolution: $1920 \times 1080$ or $1280 \times 960$) mounted on a custom delta-wing airframe. As a preprocessing, the Purdue dataset masks out the pitot tube region which is not moving in the videos. For each video, there are multiple target UAV (up to four) which have various appearances and shapes.

In the Purdue dataset, UAV and the environment is single, therefore not suitable for a UAV detection task, and instead more suitable for small target UAV tracking problem research.

### 2.2.2. Flying Objects Dataset

The study by Rozantsev et al. [29] including 20 video sequences, each with an average of 4000 images of $752 \times 480$. The Flying Objects dataset were acquired by a camera mounted on a drone filming similar videos while flying indoors and outdoors. The outdoor sequences present a broad variety of lighting and weather conditions. All these videos contain up to two objects of the same category per frame. However, the shape of the drones is rarely perfectly visible, and thus their appearance is extremely variable due to changing altitudes, lighting conditions, and even aliasing and color saturation due to their small apparent sizes.

The Flying Objects dataset is all gray-scale and is suitable for studying how to track fast moving targets, rather than MOT tasks.

### 2.2.3. Anti-UAV Dataset

Jiang et al. [30] collected 318 RGB-T video pairs, each containing an RGB video and a thermal video. The Anti-UAV dataset records various videos of several UAV types flying in the air. To ensure the diversity of data, UAV, mainly from DJI and Parrot, are utilized to collect tracking data. The videos recorded include two lighting conditions (day and night), two light modes (infrared and visible) and diverse backgrounds (buildings, cloud, trees, etc.). Each video is stored in an MP4 file with a frame rate of 25 FPS.

The Anti-UAV dataset contains RGB data and infrared data, which is convenient for the study of multi-modal fusion tracking. However, the problem is that the shooting environment is single, and the infrared camera and RGB camera are not aligned in time and space.

### 2.2.4. Drone-vs-Bird Dataset

In [31], the training data consists of a collection of 11 MPEG4-coded static-camera videos where a drone enters the scene at some point. Birds or other scene elements are not annotated. The Drone-vs-Bird dataset is increased by the need to cope with very diverse background and illumination conditions, as well as with different scales (zoom), viewpoints, low contrast, and the presence of birds.

The Drone-vs-Bird dataset includes not only abundant UAV and environmental data, but also some bird data. When the UAV is far away, it is like birds in appearance. Therefore, the emergence of this dataset can help researchers to study the problem of remote UAV and bird identification.

Whether it is an image-based UAV dataset or video-based UAV dataset, most UAV datasets are single or a few UAV objects, and most of them are either private or have only a small amount of data. Datasets suitable for UAV swarm detecting and tracking have not been reported. In general, the detection and tracking of UAV swarm objects is faced with more complex challenges than the detection and tracking of single or a few UAV objects. To solve these problems, this study constructs a dataset, named the UAVSwarm dataset; 13 different scenes and more than 19 types of UAV were recorded, including 12,598 annotated images, and the number of UAV in each sequence is 3 to 23. There are abundant scale variations, perspective transformations, and complex backgrounds for UAV targets in the UAVSwarm dataset. The UAVSwarm dataset can be applied to both UAV detection tasks and UAV swarm MOT tasks. The detailed description is presented in Section 3.

### 2.3. UAV Detection

Computer vision technology has been applied in the field of UAV since the 1990 s. In the early years, it was limited by the poor computing ability of microprocessors. Although the related algorithms have some optimization, the overall development is slow. After decades of development, with the significant improvement of processor computing power, computer vision technology to solve the problems in the application of UAV is more useful. Although the processing speed of UAV hardware has been greatly improved at present, the detection speed is still one of the key directions of research. In addition, it is affected by the low target pixels and complex environmental factors during aerial photography.

During research on detection speed, SlimYOLOv3 [32] proposed an improved version of YOLOv3. SlimYOLOv3 pruned the execution channel of the convolution layer of the original model. The evaluation on the VisDrone2018-Det benchmark dataset showed that the parameter size and the floating-point operation had a significant decrease, and the running speed had been successfully improved by about twice, and the detection accuracy was like that of YOLOv3 [33]. Excessive noise information in UAV images under complex background [34] use RPN to suppress noise information. In the aspect of small target detection, to improve the accuracy of detection, ref. [35,36] propose the FPN algorithm.

The related research of the target detection algorithm in the field of UAV has had considerable attention. Although the related algorithm optimization has achieved good results, there is still room for optimization and improvement. Judging from the main

research directions in recent years, it mainly focuses on seeking the best fit point between detection accuracy and detection speed. Although the target detection based on the YOLO algorithm is not long, the research trend is more in line with the development of UAV.

*2.4. Multiple Object Tracking*

Multiple object tracking (MOT) has attracted the considerable attention of researchers because of its great commercial potential. Most MOT algorithms include four steps: detection, feature extraction and motion prediction, similarity calculation, and data association. MOT needs to track multiple targets at the same time. The interaction between different targets, occlusion, similar appearance, in-out view and so on have brought great challenges to MOT tasks. At present, the mainstream research methods are to mark the detection box based on the target detection, and then deal with the data association problem.

SORT is mainly composed of a target detection module and data association module. SORT has low complexity, a simple frame structure and a fast running speed. Although the data association model is relatively simple and easily occluded, it paves the way for the subsequent proposal of many excellent algorithms. DeepSORT is an improvement of SORT, which can better deal with the situation that the target is occluded for a long time. The MOTA value increases little, but the IDswitch frequency is reduced by 45%, and the speed is close to the real-time requirement (20 FPS). MOTDT [37] improves the category occlusion problem in crowded scenes and uses the Re-ID feature of deep learning as the reference of appearance model to enhance the recognition ability. Moreover, the computational complexity of the algorithm is low, and the running speed can reach 20.6 FPS. JDE, based on YOLOv3 and MOTDT, achievea end-to-end visual MOT, this fusion strategy has a high-speed advantage in crowded or complex scenes. FairMot is compared with JDE as a one-shot MOT system; the MOTA value and speed are significantly improved, and the problem of high IDswitch is also suppressed, which can fully meet the real-time requirements.

In recent years, the development of the MOT algorithm based on deep learning can be seen that the existing MOT objects are often pedestrians, vehicles, and birds. MOT algorithm based on UAV swarm has not been studied yet. For the security requirements of low-altitude airspace defense and anti-UAV strategy, it is more urgent and practical to study the UAV swarm detection and tracking tasks. In addition, compared with pedestrians, UAV are a smaller target, with faster moving speed and more obvious scale transformation. Research of the MOT object being an UAV swarm is more challenging than that of the MOT object being a pedestrian.

**3. Materials and Methods**

Many studies have shown that datasets are essential for the training and testing of MOT models. Therefore, a well-labeled UAV swarm dataset for detecting and tracking UAV swarm tasks is established, named the UAVSwarm dataset, which provides data support for the training and testing of subsequent UAV detection tasks and UAV swarm MOT tasks. In this study, 72 UAV swarm image sequences were collected and processed, and the UAV appearing in each sequence were manually annotated according to a clear protocol. A new dataset containing 12,598 UAV swarm images was constructed; among them, the maximum number of UAV in a frame of image is 23. To train and test the MOT model of the UAV swarm, 72 sequences in the dataset were divided into a training set and a testing set. The training set contains 36 sequences and 6844 images, and the testing set contains 36 sequences and 5754 images.

The UAVSwarm dataset includes both ground-to-air UAV swarm and air-to-ground UAV swarm, which makes the background have complex and dynamic changes in the sky, ground, sky and ground, as well as background and light. Each sequence in this dataset contains dozens to dozens of UAV objects. There are some motion modes in the image sequence, such as leaving the field of vision, entering the field of vision, formation transformation and fast motion. The cameras shooting UAV swarm in this dataset have both

static and moving states. Therefore, this dataset contains many realistic challenges faced by UAV detection tasks and UAV swarm MOT tasks. For example, cloudy clouds and strong light make it difficult to identify UAV, tree occlusion will lead to tracking interruption, and rapid movement of small target UAV will lead to frequent ID switch. Figures 1 and 2 show the sample images of the training set and the testing set in the UAVSwarm dataset, respectively.

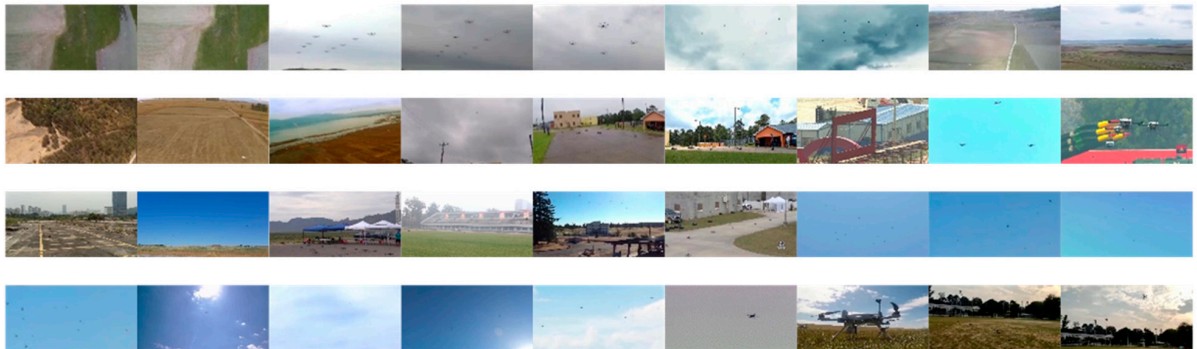

**Figure 1.** Sample images of the UAVSwarm dataset training sequences.

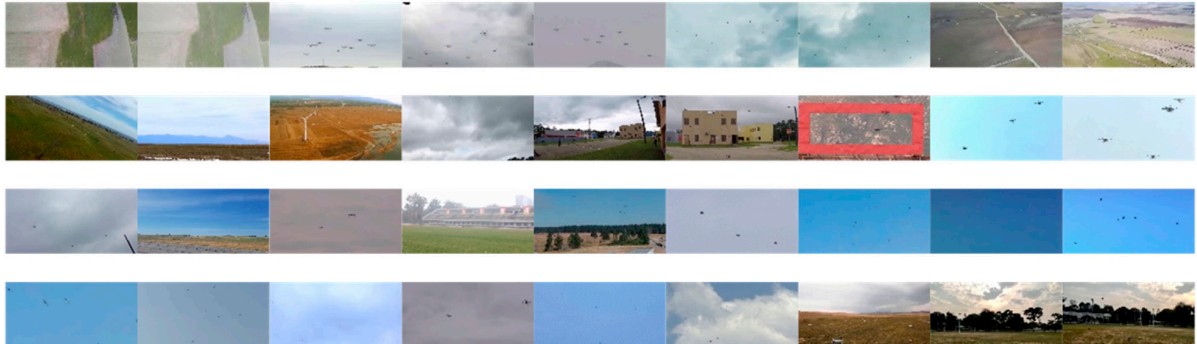

**Figure 2.** Sample images of the UAVSwarm dataset testing sequences.

*3.1. Annotation Rules*

A set of rules are followed, and a bounding box is used to label each moving UAV in each sequence as accurately as possible. In the following, a clear protocol that ensures consistency throughout the dataset is defined. Sample annotated images of the UAVSwarm dataset are shown in Figure 3.

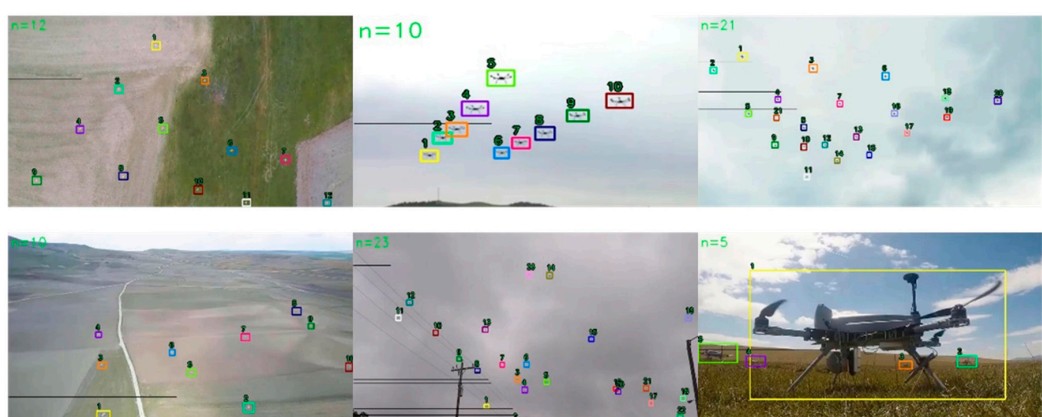

**Figure 3.** Sample annotated images of the UAVSwarm dataset.

### 3.1.1. Object Class

In the process of constructing the UAVSwarm dataset, MOT tasks of the UAV swarm are the focus of this study. Therefore, to ensure the consistency of UAVSwarm dataset image annotation, this study strictly follows the following image annotation rules:

1.　In each sequence, the UAV object is marked as early as possible and ended as late as possible. In other words, if the UAV object is in the field of vision and its path can be clearly determined, ID can be retained;
2.　In each frame, the UAV object of all types and all poses are labeled, and some UAV object images are shown in Figure 4;
3.　In each frame, the bounding box of the object should contain all the pixels belonging to this UAV object, and the bounding box should be as close as possible to the UAV object;
4.　If the exact location of the UAV object can be specified, always comment in the sequence. If the occlusion is very long and simple reasoning (e.g., constant velocity assumption) cannot be used to determine the path of the UAV object, then a new ID will be assigned after the UAV object reappears.

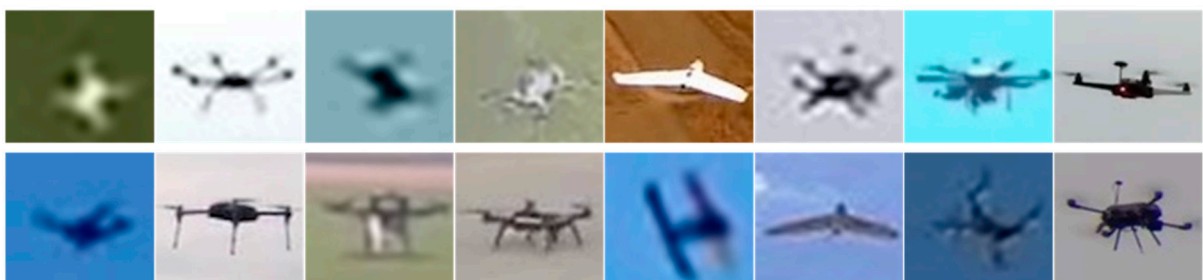

**Figure 4.** Sample images of some UAV in the UAVSwarm dataset.

### 3.1.2. Bounding Box Alignment

During the construction of the UAVSwarm dataset, the bounding box of the object is aligned as accurately as possible with the rectangular domain occupied by the UAV. The bounding box should contain all the pixels belonging to the UAV object and be as close as possible so that no object pixels should be left outside the border. If the UAV is partially occluded, the rectangular domain occupied by the UAV is estimated according to other available information (such as predicted size, shadow, reflection, previous and future frames, and other clues). If an UAV is clipped by an image border, the object border is estimated to exceed the original frame to represent the entire UAV and to estimate the clip level of the UAV object.

### 3.1.3. Start and End of Trajectories

In the construction process of the UAVSwarm dataset, if the position of the UAV and the occupied rectangular domain can be accurately determined, the bounding box (trajectory) will appear. This usually occurs in the following cases: ≈50% of UAV become visible. Similarly, when no precise positioning is possible, the trajectory ends. In other words, to ensure accuracy, UAV objects should be marked as early as possible and ended as late as possible. The bounding box coordinates may exceed the visible area. If a UAV leaves the field of vision and appears later, they will be assigned a new ID.

### 3.2. Dataset Sequences Information

In this study, 72 UAV swarm image sequences were collected and processed. Among them, the shortest sequence contained 58 images, and the longest sequence contained 705 images. The 72 sequences in the dataset are divided into a training set and testing set. The training set contains 36 sequences and 6844 images, and the testing set contains

36 sequences and 5754 images. Tables 1 and 2 summarize the frame rate, image size, sequence length, and camera movement of the training sequence and testing sequence in the dataset. At the same time, Tables 3 and 4, respectively, count the number of the bounding box in the training sequence and testing sequence in the dataset, as well as the minimum height and maximum height of the bounding box.

**Table 1.** Overview of the training sequences in the UAVSwarm dataset.

| Name | FPS | Resolution | Length | Camera |
|---|---|---|---|---|
| UAVSwarm-01 | 30 | 812 × 428 | 164 | static |
| UAVSwarm-03 | 30 | 812 × 428 | 150 | static |
| UAVSwarm-05 | 30 | 446 × 270 | 153 | static |
| UAVSwarm-07 | 30 | 446 × 270 | 141 | static |
| UAVSwarm-09 | 30 | 446 × 270 | 68 | static |
| UAVSwarm-11 | 30 | 812 × 428 | 160 | static |
| UAVSwarm-13 | 30 | 812 × 428 | 119 | static |
| UAVSwarm-15 | 30 | 812 × 425 | 116 | moving |
| UAVSwarm-17 | 30 | 812 × 428 | 100 | static |
| UAVSwarm-19 | 30 | 639 × 328 | 205 | moving |
| UAVSwarm-21 | 30 | 847 × 412 | 93 | static |
| UAVSwarm-23 | 30 | 799 × 477 | 73 | moving |
| UAVSwarm-25 | 30 | 863 × 411 | 244 | moving |
| UAVSwarm-27 | 30 | 863 × 480 | 193 | moving |
| UAVSwarm-29 | 30 | 863 × 407 | 166 | static |
| UAVSwarm-31 | 30 | 863 × 441 | 204 | moving |
| UAVSwarm-33 | 30 | 1279 × 630 | 193 | static |
| UAVSwarm-35 | 30 | 1279 × 621 | 96 | moving |
| UAVSwarm-37 | 30 | 1279 × 606 | 330 | moving |
| UAVSwarm-39 | 30 | 1919 × 1079 | 483 | moving |
| UAVSwarm-41 | 30 | 847 × 478 | 181 | static |
| UAVSwarm-43 | 30 | 1919 × 870 | 408 | static |
| UAVSwarm-45 | 30 | 1279 × 711 | 219 | moving |
| UAVSwarm-47 | 30 | 1279 × 618 | 234 | moving |
| UAVSwarm-49 | 30 | 863 × 378 | 195 | static |
| UAVSwarm-51 | 30 | 625 × 292 | 152 | static |
| UAVSwarm-53 | 30 | 625 × 292 | 117 | moving |
| UAVSwarm-55 | 30 | 625 × 292 | 98 | static |
| UAVSwarm-57 | 30 | 639 × 292 | 146 | static |
| UAVSwarm-59 | 30 | 844 × 344 | 65 | moving |
| UAVSwarm-61 | 30 | 1279 × 655 | 300 | static |
| UAVSwarm-63 | 30 | 807 × 424 | 120 | moving |
| UAVSwarm-65 | 30 | 863 × 485 | 74 | static |
| UAVSwarm-67 | 30 | 813 × 427 | 98 | static |
| UAVSwarm-69 | 30 | 640 × 352 | 333 | static |
| UAVSwarm-71 | 30 | 640 × 352 | 653 | moving |
| Total training | | | 6844 | |

**Table 2.** Overview of the testing sequences in the UAVSwarm dataset.

| Name | FPS | Resolution | Length | Camera |
|---|---|---|---|---|
| UAVSwarm-02 | 30 | 812 × 428 | 156 | static |
| UAVSwarm-04 | 30 | 812 × 428 | 161 | static |
| UAVSwarm-06 | 30 | 446 × 270 | 76 | static |
| UAVSwarm-08 | 30 | 446 × 270 | 115 | static |
| UAVSwarm-10 | 30 | 863 × 467 | 58 | static |

**Table 2.** *Cont.*

| Name | FPS | Resolution | Length | Camera |
|---|---|---|---|---|
| UAVSwarm-12 | 30 | 812 × 428 | 116 | static |
| UAVSwarm-14 | 30 | 812 × 428 | 115 | static |
| UAVSwarm-16 | 30 | 812 × 428 | 65 | moving |
| UAVSwarm-18 | 30 | 625 × 291 | 84 | static |
| UAVSwarm-20 | 30 | 720 × 479 | 62 | moving |
| UAVSwarm-22 | 30 | 764 × 479 | 72 | moving |
| UAVSwarm-24 | 30 | 844 × 455 | 72 | moving |
| UAVSwarm-26 | 30 | 863 × 364 | 126 | static |
| UAVSwarm-28 | 30 | 810 × 475 | 131 | moving |
| UAVSwarm-30 | 30 | 863 × 472 | 105 | static |
| UAVSwarm-32 | 30 | 863 × 467 | 76 | static |
| UAVSwarm-34 | 30 | 1279 × 630 | 176 | static |
| UAVSwarm-36 | 30 | 1279 × 625 | 81 | static |
| UAVSwarm-38 | 30 | 1279 × 606 | 452 | moving |
| UAVSwarm-40 | 30 | 1919 × 1079 | 448 | static |
| UAVSwarm-42 | 30 | 847 × 424 | 120 | static |
| UAVSwarm-44 | 30 | 1919 × 870 | 353 | static |
| UAVSwarm-46 | 30 | 1279 × 684 | 238 | static |
| UAVSwarm-48 | 30 | 1279 × 598 | 221 | static |
| UAVSwarm-50 | 30 | 625 × 292 | 170 | static |
| UAVSwarm-52 | 30 | 625 × 291 | 157 | moving |
| UAVSwarm-54 | 30 | 625 × 292 | 110 | moving |
| UAVSwarm-56 | 30 | 625 × 292 | 89 | static |
| UAVSwarm-58 | 30 | 863 × 465 | 122 | static |
| UAVSwarm-60 | 30 | 846 × 341 | 68 | static |
| UAVSwarm-62 | 30 | 847 × 474 | 140 | static |
| UAVSwarm-64 | 30 | 863 × 378 | 75 | static |
| UAVSwarm-66 | 30 | 863 × 464 | 73 | moving |
| UAVSwarm-68 | 30 | 811 × 426 | 59 | moving |
| UAVSwarm-70 | 30 | 639 × 351 | 307 | static |
| UAVSwarm-72 | 30 | 639 × 351 | 705 | moving |
| Total testing | | | 5754 | |

**Table 3.** The bounding box information of the training sequence in the UAVSwarm dataset.

| Name | Total of Bounding Boxes | Min Height | Max Height |
|---|---|---|---|
| UAVSwarm-01 | 2984 | 10 | 23 |
| UAVSwarm-03 | 1354 | 10 | 24 |
| UAVSwarm-05 | 1530 | 16 | 27 |
| UAVSwarm-07 | 1394 | 10 | 38 |
| UAVSwarm-09 | 272 | 20 | 37 |
| UAVSwarm-11 | 3356 | 10 | 20 |
| UAVSwarm-13 | 2499 | 10 | 22 |
| UAVSwarm-15 | 1286 | 10 | 30 |
| UAVSwarm-17 | 669 | 10 | 21 |
| UAVSwarm-19 | 1217 | 10 | 24 |
| UAVSwarm-21 | 206 | 13 | 30 |
| UAVSwarm-23 | 494 | 12 | 48 |
| UAVSwarm-25 | 4091 | 10 | 28 |
| UAVSwarm-27 | 772 | 32 | 18 |
| UAVSwarm-29 | 1151 | 12 | 24 |
| UAVSwarm-31 | 321 | 28 | 50 |
| UAVSwarm-33 | 733 | 15 | 54 |
| UAVSwarm-35 | 384 | 78 | 161 |
| UAVSwarm-37 | 1251 | 17 | 53 |

**Table 3.** *Cont.*

| Name | Total of Bounding Boxes | Min Height | Max Height |
|---|---|---|---|
| UAVSwarm-39 | 3482 | 14 | 26 |
| UAVSwarm-41 | 1601 | 14 | 58 |
| UAVSwarm-43 | 1632 | 23 | 85 |
| UAVSwarm-45 | 1967 | 10 | 20 |
| UAVSwarm-47 | 1423 | 15 | 45 |
| UAVSwarm-49 | 2145 | 10 | 15 |
| UAVSwarm-51 | 1336 | 10 | 23 |
| UAVSwarm-53 | 663 | 10 | 26 |
| UAVSwarm-55 | 1294 | 10 | 26 |
| UAVSwarm-57 | 261 | 10 | 29 |
| UAVSwarm-59 | 322 | 10 | 18 |
| UAVSwarm-61 | 3543 | 10 | 45 |
| UAVSwarm-63 | 1080 | 12 | 21 |
| UAVSwarm-65 | 125 | 43 | 60 |
| UAVSwarm-67 | 443 | 15 | 331 |
| UAVSwarm-69 | 913 | 15 | 29 |
| UAVSwarm-71 | 1467 | 15 | 34 |

**Table 4.** The bounding box information of the testing sequence in the UAVSwarm dataset.

| Name | Total of Bounding Boxes | Min Height | Max Height |
|---|---|---|---|
| UAVSwarm-02 | 2983 | 10 | 21 |
| UAVSwarm-04 | 3034 | 11 | 22 |
| UAVSwarm-06 | 760 | 14 | 24 |
| UAVSwarm-08 | 1090 | 14 | 39 |
| UAVSwarm-10 | 406 | 27 | 35 |
| UAVSwarm-12 | 2320 | 11 | 22 |
| UAVSwarm-14 | 1886 | 10 | 26 |
| UAVSwarm-16 | 721 | 12 | 30 |
| UAVSwarm-18 | 647 | 15 | 30 |
| UAVSwarm-20 | 304 | 14 | 26 |
| UAVSwarm-22 | 228 | 32 | 54 |
| UAVSwarm-24 | 500 | 10 | 26 |
| UAVSwarm-26 | 1488 | 10 | 11 |
| UAVSwarm-28 | 873 | 11 | 24 |
| UAVSwarm-30 | 486 | 10 | 29 |
| UAVSwarm-32 | 320 | 17 | 54 |
| UAVSwarm-34 | 528 | 52 | 77 |
| UAVSwarm-36 | 247 | 57 | 79 |
| UAVSwarm-38 | 1649 | 11 | 42 |
| UAVSwarm-40 | 3604 | 12 | 38 |
| UAVSwarm-42 | 347 | 14 | 43 |
| UAVSwarm-44 | 1308 | 30 | 95 |
| UAVSwarm-46 | 2670 | 11 | 45 |
| UAVSwarm-48 | 663 | 35 | 47 |
| UAVSwarm-50 | 1868 | 11 | 20 |
| UAVSwarm-52 | 750 | 10 | 22 |
| UAVSwarm-54 | 665 | 10 | 27 |
| UAVSwarm-56 | 378 | 10 | 34 |
| UAVSwarm-58 | 745 | 13 | 34 |
| UAVSwarm-60 | 314 | 10 | 20 |
| UAVSwarm-62 | 435 | 14 | 49 |
| UAVSwarm-64 | 396 | 10 | 17 |
| UAVSwarm-66 | 219 | 19 | 26 |
| UAVSwarm-68 | 976 | 10 | 80 |
| UAVSwarm-70 | 789 | 12 | 30 |
| UAVSwarm-72 | 1350 | 15 | 50 |

## *3.3. Data Format*

Since the MOT16 [38] dataset is used in the Computer Vision and Pattern Recognition 2019 (CVPR,2019) tracking challenge of the CVPR and becomes an authoritative dataset for

single object track (SOT) tasks and MOT tasks, many existing tracking models are evaluated on the dataset. Therefore, the data format of the UAVSwarm dataset also uses the dataset format in MOT16. At the same time, all images are in JPEG format and are named as the file name of six digits (e.g., 000001.jpg).

### 3.3.1. Detection Files

To focus on the tracking performance of UAV swarm MOT tasks, test results of excellent YOLOX detectors are used as detection files. The mAP and recall results of the UAVSwarm dataset detected by YOLOX are shown in Figure 5. The detection files are simple comma separated value (CSV) files. Each row represents an object, and each row contains nine values. The first number indicates which frame the object appears in. The second number represents the ID of the object (set to '−1' in the detection files, because the ID has not been assigned), by assigning a unique ID, each object can only be specified to a trajectory. The next four numbers represent the position of the UAV bounding box in the two-dimensional image coordinates, representing the upper left corner coordinates x, y and the width and height of the bounding box. The seventh number represents its confidence score. The last three digits of the detection files are represented as the ignored state (set to '−1'). Examples of detection files are as follows:

1, −1, 174, 243, 12, 12, 1, −1, −1, −1
1, −1, 215, 326, 13, 14, 1, −1, −1, −1
1, −1, 235, 167, 13, 14, 1, −1, −1, −1
1, −1, 273, 250, 11, 12, 1, −1, −1, −1

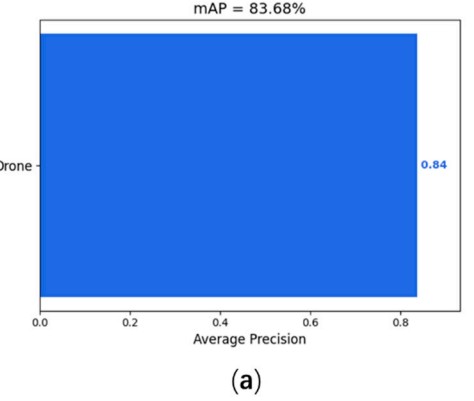
(**a**)

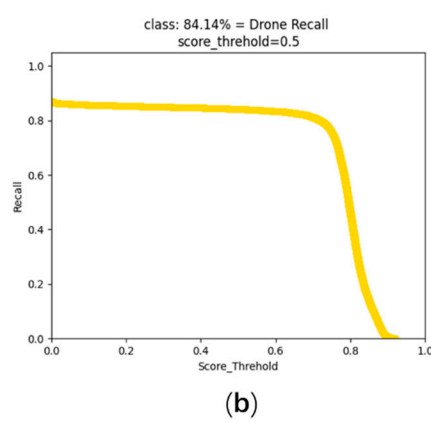
(**b**)

**Figure 5.** (**a**) The mAP results of the UAVSwarm dataset detected by YOLOX. (**b**) The recall results of the UAVSwarm dataset detected by YOLOX.

### 3.3.2. Annotation Files

In the UAVSwarm dataset, the annotation files are simple comma separated value (CSV) files. Each row represents an object, and each row contains nine values. The first six digital meanings are the same as the first six digital meanings of the test files. The seventh number represents the confidence score and serves as a symbol of whether input is considered ('0' means that this object is ignored in calculation, '1' means that this object is marked as an activity, and this value is set to '1' in annotation files). The eighth number represents the type of annotated object (since there is only one UAV category in this dataset, the values are set to '1'). The ninth number represents the visibility ratio of each bounding box, ranging from 0 to 1, which is judged by the occlusion degree of another static or moving object or by the clipping of image boundary (the value in annotation files is set to '1'). Examples of annotation files are as follows:

1, 1, 352, 20, 11, 11, 1, 1, 1
2, 1, 352, 20, 11, 11, 1, 1, 1
3, 1, 352, 20, 11, 11, 1, 1, 1
4, 1, 352, 20, 11, 11, 1, 1, 1

In addition, to obtain the effective results of the entire benchmark test, this study creates a separate CSV file for each sequence that follows the above format, named 'Sequence-Name. txt'.

## 4. Results

### 4.1. Experiment Setting and Evaluation Metrics

To test and verify the availability, consistency, and generality of the UAVSwarm dataset, this study applies the dataset to the UAV detection tasks of the Faster R-CNN and YOLOX [39], and the UAV swarm MOT tasks of the GNMOT [40] and ByteTrack [41], and uses the standards defined in Reference [42] to evaluate the performance of the UAV detection tasks and UAV swarm MOT tasks. The evaluation metrics are mainly as follows:

mAP (↑): mean average precision. A widely used evaluation metric in the object detection because of the ability to measure the performance of the localization and classification.

FPS (↑): frames per second. To measure the speed of models.

MOTA (↑): multiple object tracking accuracy. The accuracy in determining the number of objects and related attributes of the object is used to count the error accumulation in tracking, including the total number of false positives (FP), the total number of false negatives (FN), and the total number of identity switches (IDSW).

$$MOTA = 1 - \frac{\sum_t (FN_t + FP_t + IDSW_t)}{\sum_t GT_t} \tag{1}$$

Among them, $GT_t$ represents the number of ground truth in the t frame, $FN_t$ represents the number of false negatives in the t frame, $FP_t$ represents the number of false positive in the t frame, and $IDSW_t$ represents the number of identity switch in the t frame.

MOTP (↑): multiple object tracking precision. MOTP mainly quantizes the positioning accuracy of the detector, and almost does not contain information related to the actual performance of the tracker.

$$MOTP = \frac{\sum_{t,i} d_{t,i}}{\sum_t c_t} \tag{2}$$

where $c_t$ represents the number of detection boxes successfully matched with ground truth in frame t, and dt,i represents the distance measurement between matching pairs.

IDF1 (↑): The ratio of correct recognition detection to average true number and calculated detection number. To measure whether a tracker tracks an object, if possible, that is, the quality of data association.

$$IDF1 = \frac{2IDTP}{2IDTP + IDFP + IDFN} \tag{3}$$

Among them, IDTP and IDFP represent true positive ID number and false positive ID number, respectively, like P in the confusion matrix, but now it is the calculation of ID recognition accuracy; IDFN is the false negative ID number.

MT (↑): mostly tracked objects. The ratio of ground-truth trajectories that are covered by a track hypothesis for at least 80% of their respective life span.

ML (↓): mostly lost objects. The ratio of ground-truth trajectories that are covered by a track hypothesis for at most 20% of their respective life span.

FP (↓): the total number of false positives.

FN (↓): the total number of false negatives (missed objects).

Frag (↓): the total number of times a trajectory is fragmented (i.e., interrupted during tracking).

IDswitch (↓): the total number of identity switches.

In the above metrics, ↓ means that the greater the index, the better the performance, and ↓ means that the smaller the index, the better the performance.

All the experiments are carried out on a computer with 16-GB memory, Intel Corei7-10700 CPU and NVIDIA RTX2080Ti GPU. Faster R-CNN, YOLOX, GNMOT and ByteTrack

models adopt public codes, and all codes are trained using the default parameter set recommended by the authors.

### 4.2. Experimental Results and Analysis

### 4.2.1. UAV Detection

Most object detection methods include two parts: (1) a backbone model as the feature extractor to extract feature from images, and (2) a detection head as the detector to find out the classification and localization feature of objects. Nowadays, most object detection models are based on CNNs [43] to extract feature of inputs. In this study, the advanced CNN model is integrated, with Resnet-50 [44] and Darknet-53 [45] as the backbone of these object detect methods, and Faster R-CNN and YOLOX to extract UAV features in images. The baseline models are listed in Table 5.

**Table 5.** Baseline object detection methods of UAV detection.

| Method | Backbone | Detection Head |
|:---:|:---:|:---:|
| Faster R-CNN | Resnet-50 | Two-stage |
| YOLOX | DarkNet-53 | One-stage |

A detector is trained to localize and identify more than 19 types of UAV in the UAVSwarm dataset. A total of 6844 images of UAV were used to train the detector and evaluate on the testing sequence of the UAVSwarm dataset. For speed of detection, despite different feature extractors, the FPS results of the YOLOX is always the best compared with Faster R-CNN UAV detection. For performance of detection, YOLOX has the best performance for Darknet-53 in the UAV detection task, and mAP reached 83.68. The results of Faster R-CNN and YOLOX on the testing sequence of the UAVSwarm dataset are listed in Table 6.

**Table 6.** Results of UAV detection.

| Detection Head | Backbone | Input Size | Speed (FPS) | mAP (%) |
|:---:|:---:|:---:|:---:|:---:|
| Faster R-CNN | Resnet-50 | $640 \times 640$ | 17 | 50.75 |
| YOLOX | DarkNet-53 | $640 \times 640$ | 119 | 83.68 |

To visualize the effects of the two detectors more intuitively, 24, 30, 40, and 68 testing sequences of the same frame image were compared; the visualization results are shown in Figure 6.

All in all, on the UAV detection task, whichever model has the best results in terms of speed and accuracy, these widely recognized object detection methods have a reasonable and accurate result on our dataset. This is enough to show that our dataset provides data support for UAV detection. In addition, the CNN models of these detection methods have indeed helped the baselines to achieve better results in speed or mAP.

### 4.2.2. UAV Swarm MOT

GNMOT is a new near online MOT method with an end-to-end graph network. Specifically, GNMOT designs an appearance graph network and a motion graph network to capture the appearance and the motion similarity separately. The updating mechanism is carefully designed in the graph network, which means that nodes, edges, and the global variable in the graph can be updated. The global variable can capture the global relationship to help tracking. Finally, a strategy to handle missing detections is proposed to remedy the defect of the detectors.

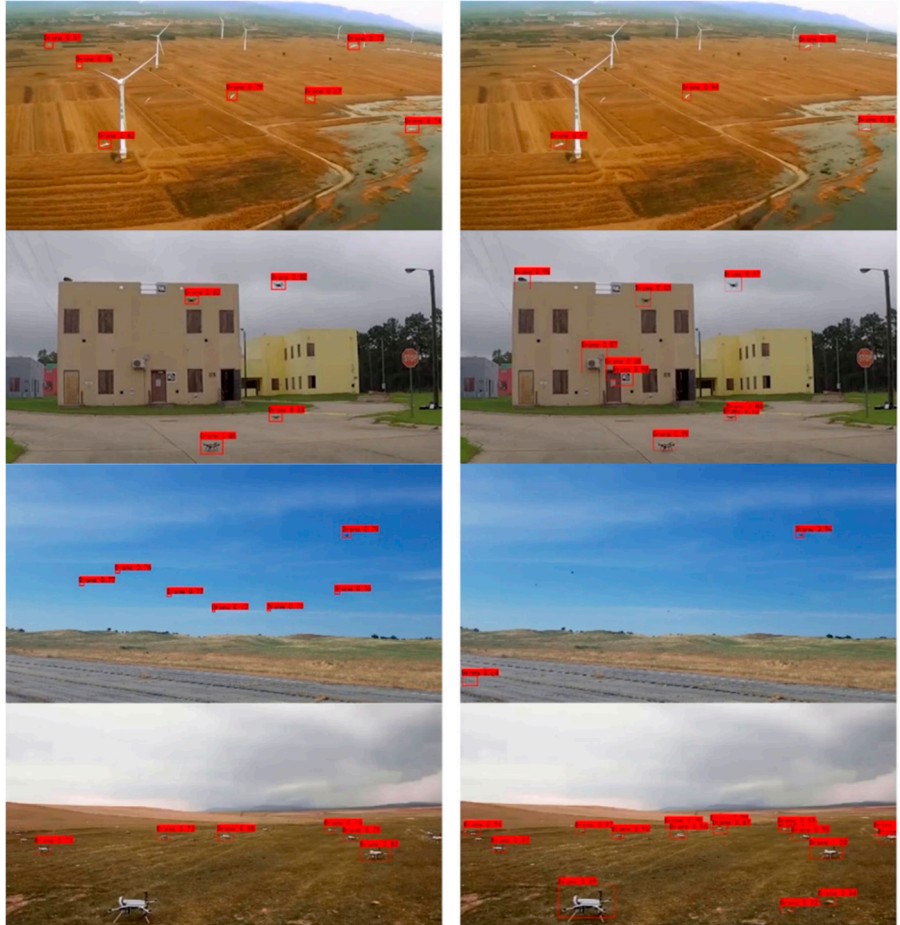

**Figure 6.** Sample visualization results images of the testing sequence (Right: Faster R-CNN; Left: YOLOX).

The GNMOT model is applied to UAV swarm MOT tasks. To focus on evaluating the tracking performance of GNMOT, an annotation file that removes ID information is used as a detection file. Then, experiments were carried out in 36 testing sequences in the UAVSwarm dataset. The experimental results of evaluation indexes IDF1 and MOTA are shown in Figure 7. The experimental results show that MOTA and IDF1 of 25 testing sequences reaches 100%. However, sequences 28,70, and 72 have nearly 100% MOTA, but IDF1 does not reach 80%; this is because the MOT tasks introduce the ID information, which will pay more attention to whether the ID of the initial trajectory created by the tracker can be 'from one to the end '. If the ID switch is too early, the final trajectory ID must be far from the initial trajectory ID when the number of IDswitch is the same; this makes the IDF1 score lower. So, measuring whether a model is suitable for UAV swarm MOT tasks requires not only excellent small target detectors, but also a tracker that can accurately match the trajectory.

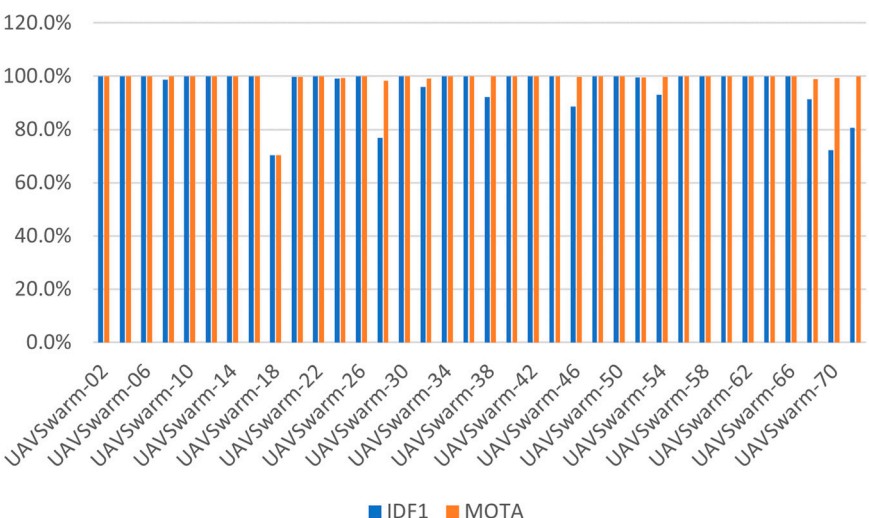

**Figure 7.** Comparison of IDF1 and MOTA results on the UAVSwarm dataset testing set using GNMOT.

ByteTrack is a tracking method based on tracking-by-detection, and it proposes a simple and efficient data association method BYTE. The biggest difference between it and the previous tracking model is that it is not simply to remove the low-score detection results. By using the similarity between the detection box and the tracking trajectory, the background is removed from the low-resolution detection results while retaining the high-resolution detection results, and the real objects (occlusion, blur, and other difficult samples) are excavated, to reduce the missed detection and improve the coherence of the trajectory. However, it should be noted that since ByteTrack does not use the appearance feature to match, the tracking effect is very dependent on the detection effect. If the effect of the detector is good, the tracking will also achieve good results; however, if the detection effect is not good, it will seriously affect the tracking effect. At present, as the state-of-the-art model of the MOT Challenge, the ByteTrack model uses the detector YOLOX with excellent performance to get the detection results. In the process of data association, like SORT, only Kalman filter is used to predict the position of the tracking trajectory of the current frame in the next frame. The distance metric between the predicted frame and the actual detection frame is used as the similarity of the two matching, and the matching is completed by the Hungarian algorithm.

In this study, the UAVSwarm dataset is used in ByteTrack model, and the experimental results are shown in Figure 8. The experimental results show that 13 sequences have MOTA higher than 80%, 7 sequences have negative MOTA (the MOTA can become negative when the error generated by the tracker exceeds the object in the scene), and 14 sequences have MOTA between 0% and 80%. The 13 sequences that have MOTA higher than 80% are all a simple way of movement, and the scene is relatively not complex, so it is easy for the tracker to generate the correct trajectory. The 7 sequences with negative MOTA are all complex modes of motion: UAV move fast, the UAV continue to fly into the out of the screen, and the trajectories overlap with each other. The experimental results show that even if the YOLOX detector can reach 83.68 mAP, the tracking results will also produce negative numbers due to the large scale change, fast moving speed, frequent access to the screen and trajectory overlap of the UAV. Therefore, the completion of UAV swarm MOT tasks requires not only excellent detectors, but also excellent trackers to maintain tracking consistency and avoid object jumping.

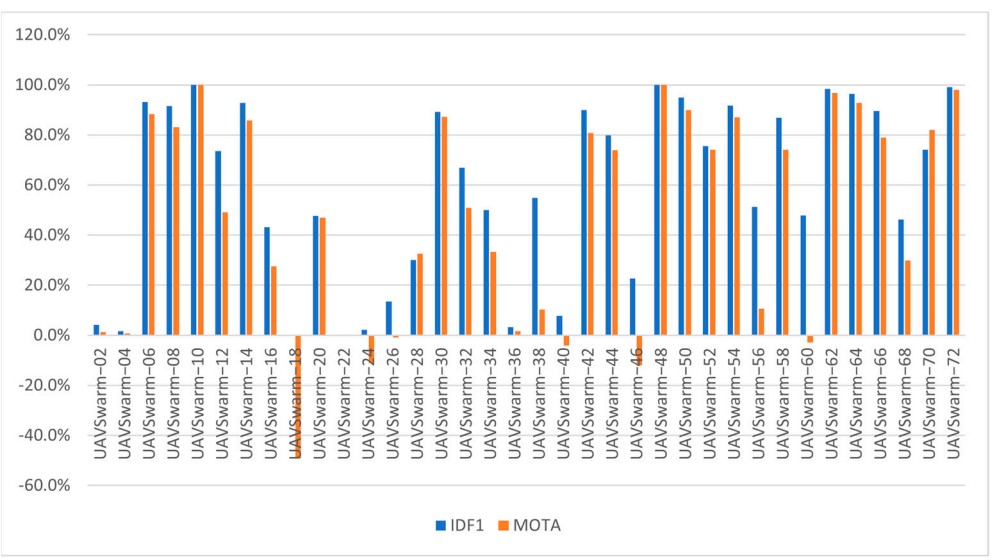

**Figure 8.** Comparison of IDF1 and MOTA results on the UAVSwarm dataset testing set using ByteTrack.

Therefore, the MOT tasks with the object of UAV swarm are very challenging. This experiment further verifies the availability and versatility of the UAVSwarm dataset constructed in this study.

### 4.3. Visual Tracking Results

To illustrate the model performance more intuitively, Figures 9–12 show the visual tracking results of the above two models on some UAVSwarm dataset videos. The detection box ID number of each sequence in the ByteTrack model is accumulated according to the ID number of the previous sequence, and the detection box ID of each sequence in GNMOT model starts from 1. The ByteTrack model uses the detector YOLOX with excellent current performance to obtain the detection results, and the GNMOT model uses the detection files provided by us (the length, width, and position of the detection box in the detection files are consistent with the bounding box of the annotation files).

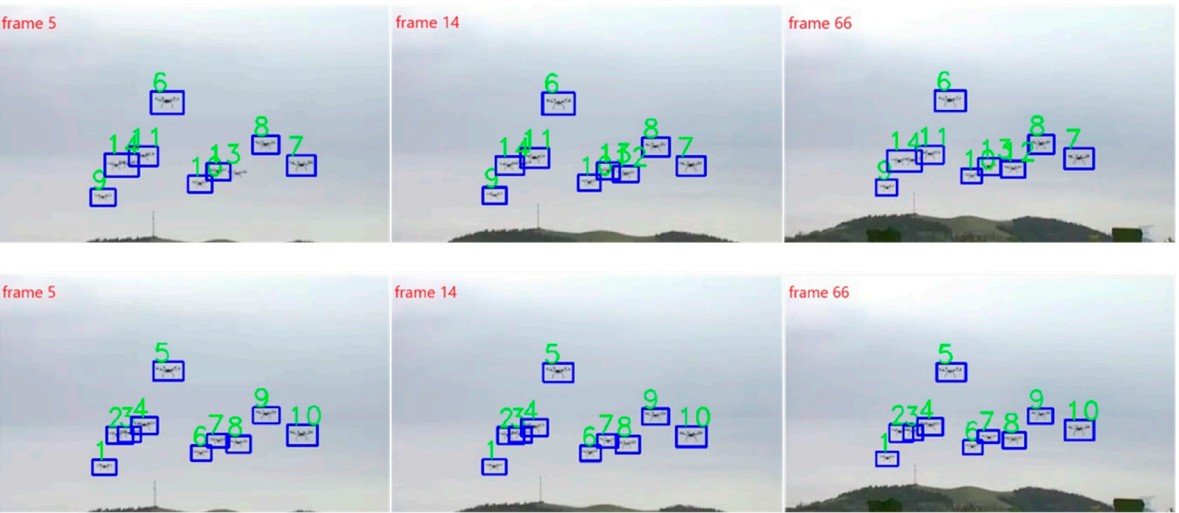

**Figure 9.** Tracking results of UAVSwarm−06 (Top: ByteTrack; Bottom: GNMOT; 5th, 14th, and 66th frames, from left to right).

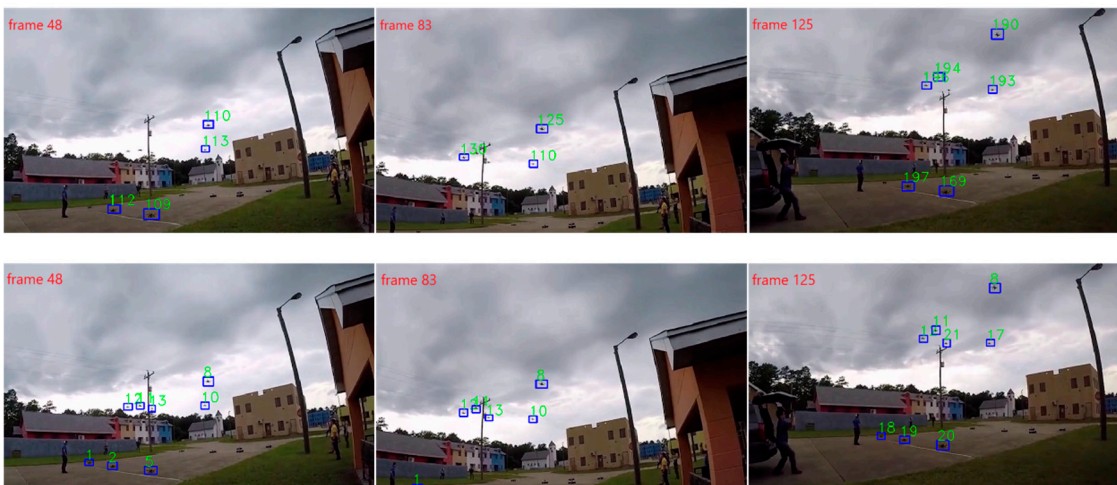

**Figure 10.** Tracking results of UAVSwarm−28 (Top: ByteTrack; Bottom: GNMOT; 48th, 83rd, and 125th frames, from left to right).

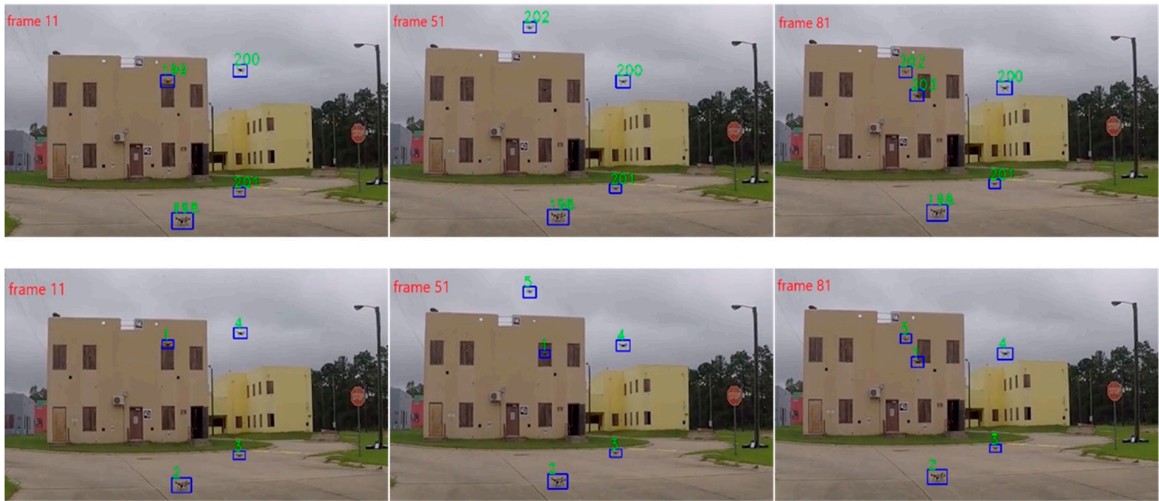

**Figure 11.** Tracking results of UAVSwarm−30 (Top: ByteTrack; Bottom: GNMOT; 11th, 51st, and 81st frames, from left to right).

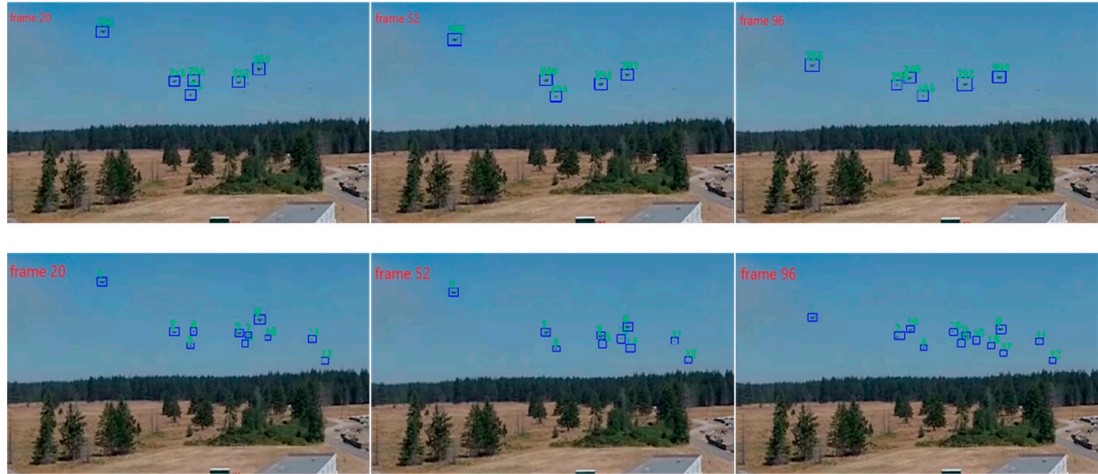

**Figure 12.** Tracking results of UAVSwarm−46 (Top: ByteTrack; Bottom: GNMOT; 20th, 52nd, and 96th frames, from left to right).

Figure 9 shows a video of UAVSwarm−06. The shooting background of this video is the sky, and the background is relatively simple and clean. However, the UAV is close to each other, and UAV are overlapped and occluded. In frame 5, the detection box of ByteTrack model ID 14 contains two UAV, and the GNMOT model uses the detecting files provided by us to distinguish two UAV that are close to each other, with IDs of 2 and 3, respectively. In the 14th frame, the ByteTrack model re-recognizes the detection box with the previous missing ID of 12. In the 66th frame, there is still no distinction between two UAV in the detection box of ByteTrack model ID 14. Although the UAV with ID number 2 and 3 of GNMOT model continues to block, it can always be accurately tracked.

Figure 10 shows a video of UAVSwarm−28. The shooting background of this video is complex, and the lens deviates. The two models have multiple ID switching because of lens jitter, and the houses in the background and the dark clouds have brought great challenges to tracking.

Figure 11 shows a video of UAVSwarm−30. The shooting background of this video is complex, but the lens does not deviate. In the 11th frame, ByteTrack and GNMOT detected four UAV. In the 51st frame, the UAV with ID of 199 in the ByteTrack model misses the detection box because it is like the window background. In the 81st frame, the UAV with ID of 199 in the ByteTrack model is re-identified, and the ID number is 203. The GNMOT model can accurately track five UAV in frame 11, 51, and 81, and the ID number has not been switched.

Figure 12 shows a video of UAVSwarm−46. The background of the video is simple, but the object is too small, occluded, and moves fast. The ByteTrack model makes more errors than objects in the scene because of the tracker, and MOTA becomes negative. Table 7 presents the tracking results of the UAVSwarm dataset testing sequence and mostly tracked tracklets (MT) represents the number of tracking trajectories that at least 80 % of the video frames of each object can be correctly tracked in the tracking process, mostly lost tracklets (ML) represents the number of tracking trajectories that at most 20 % of the video frames of each object can be correctly tracked in the tracking process.

**Table 7.** Partial UAVSwarm dataset testing sequence tracking results.

| Sequence | Tracker | MOTA | IDF1 | MT | ML | FP | FN | IDS |
|----------|---------|------|------|-----|-----|-----|------|-----|
| UAVSwarm−06 | ByteTrack | 88.3% | 93.2% | 9 | 1 | 0 | 89 | 0 |
| | GNMOT | 100.0% | 100.0% | 10 | 0 | 0 | 0 | 0 |
| UAVSwarm−28 | ByteTrack | 32.5% | 30.0% | 1 | 3 | 14 | 558 | 14 |
| | GNMOT | 98.4% | 76.9% | 8 | 0 | 0 | 0 | 14 |
| UAVSwarm−30 | ByteTrack | 87.2% | 89.2% | 4 | 0 | 1 | 60 | 0 |
| | GNMOT | 100.0% | 100.0% | 5 | 0 | 0 | 0 | 0 |
| UAVSwarm−46 | ByteTrack | −12.0% | 22.7% | 0 | 11 | 769 | 2220 | 8 |
| | GNMOT | 99.7% | 88.7% | 16 | 0 | 0 | 0 | 8 |

## 5. Discussion

### 5.1. UAV Swarm Dataset

The construction of UAV swarm dataset has the following difficulties. (1) With the development of deep learning, more and more people pay attention to the innovation of algorithms. However, because UAV are different from objects that can be seen everywhere in daily life, it is necessary to prepare UAV with different shapes to collect their data, and there are also requirements for shooting locations, which makes it difficult to obtain UAV data. (2) UAV detection has gradually become a research field with its unique problems. Compared with other targets, UAV objects have rich scale changes. The detection of far and near distance is very necessary, so it is necessary to collect UAV data at different scales. (3) UAV detection can be divided into ground-to-air detection and air-to-air detection, and the conversion of perspective brings different detection difficulties. Whether the dataset contains UAV under various attitudes is one of the factors affecting UAV detection. (4) The diversity of background is crucial to improve the generalization ability of UAV detection,

so it is necessary to collect data in different environments. The UAV data collected in a single environment will make the detection algorithm cannot be directly applied to the complex and changeable real environment.

### 5.2. UAV Detection

The primary difficulty of UAV detection task is the small pixel size of UAV in optical images. It is generally believed that the object whose pixel size is less than 1 % of the whole image is a small target. The object detection common dataset COCO defines the small object as the object whose pixel size is less than $32 \times 32$. Small UAV often occupy smaller pixels after imaging. In the optical image dataset of $1960 \times 1080$, most UAV occupy pixel sizes between $15 \times 15$ and $35 \times 35$, and contain less feature information. Therefore, special algorithms need to be designed for small target detection tasks of low altitude UAV.

### 5.3. UAV Swarm Multiple Object Tracking

MOT is an important computer vision problem, which has attracted more and more attention due to its great academic and commercial potential. Although there are many different methods to solve this problem, it is still a huge challenge due to factors such as sudden appearance changes and serious object occlusion. As far as we know, there is no broad comment on UAV swarm in the field of MOT.

### 6. Conclusions

In the paper, an image dataset for detecting and tracking UAV swarm is established, called the UAVSwarm dataset. It contains 13 different scenes and 19 types of UAV, including 12,598 annotated images, and the number of UAV in each sequence is 3 to 23. Then, our dataset is validated on two tasks: UAV detection and UAV swarm MOT. For UAV detection tasks, the widely used object detection methods, Faster R-CNN and YOLOX, are used as the baseline models. In addition, Resnet-50 and Darknet-53 are used as backbone models for UAV detection methods to extract more UAV-related image information. These models achieve high results in UAV detection tasks, which reveal that our dataset plays an essential and significant role in UAV detection. For UAV swarm MOT tasks, two of the most advanced MOT models, GNMOT and ByteTrack, are applied. The experimental results verify the availability, consistency, and versatility of the UAVSwarm dataset on IDF1 and MOTA evaluation metrics. The dataset constructed in this study will play a vital role in UAV swarm MOT tasks.

In future work, on the one hand, more types of UAV should be collected, including more information such as description and segmentation. On the other hand, more excellent UAV detection models and UAV swarm MOT models should be designed, which provides technical support for timely detection and accurate tracking of incoming UAV swarm. Finally, most current datasets (like PASCAL, ImageNet, etc.) for object detection are built to contain many images for nearly all object classes, rather than focusing on one issue. With the lack of dataset focusing on UAV detection and UAV swarm MOT, we believe that the high quality and large scale of UAVSwarm Dataset will become a new and challenging benchmark dataset for future research.

**Author Contributions:** Conceptualization, C.W.; methodology, C.W.; software, Y.S.; validation, Y.S., J.W. and T.W.; formal analysis, Q.G.; investigation, Y.S.; resources, C.W.; data curation, Y.S.; writing—original draft preparation, Y.S.; writing—review and editing, C.W. and Q.G.; visualization, Y.S.; supervision, C.W.; project administration, C.W.; funding acquisition, C.W. All authors have read and agreed to the published version of the manuscript.

**Funding:** This research was supported by the National Natural Science Foundation of China (Grant No. 61703287), the Scientific Research Program of Liaoning Provincial Education Department of China (Grant Nos. LJKZ0218 and JYT2020045), the Young and Middle-aged Science and Technology Innovation Talents Project of Shenyang of China (Grant No. RC210401), and the Liaoning Provincial Key R&D Program of China (Grant No. 2020JH2/10100045).

**Data Availability Statement:** Not applicable.

**Conflicts of Interest:** The authors declare no conflict of interest.

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
