# Peer review of "UAVSwarm Dataset: An Unmanned Aerial Vehicle Swarm Dataset for Multiple Object Tracking"

_remotesensing, doi:10.3390/rs14112601_

Round 1

Reviewer 1 Report

See review remarks attached for improvements. First of all, formulate clearly: what is the main goal of your paper?! To detect incoming (attacking) drone swarms, or to control your UAV fleet conducting cooperative flight missions?! The problem statement formulated in your paper involves few things about challenges of the low-altitude air defense, however, it does not formulated clearly what is your general purpose. It is not clear you design UAV dataset for defensive or offensive missions, it would be very useful to emphasize your standpoint, too.

Reviewer 2 Report

This manuscript presents a UAV swarm dataset for multiple object tracking. I have some questions about the effectiveness in real-world settings.

  1. The authors used some images to train and test the dataset. Does the real-world experiment also provide similar performance?
  2. What is the processing speed? Is it able to detect objects within a very short time? 
  3. How about the accuracy? Can it distinguish birds or other subjects?
  4. Can this dataset really work for real cases? Can it handle cases in which the UAVs are far away?
  5. How about the performance at night or on rainy days? Does infrared image work well?

Minor issue: Is it a mistake in the Lines 204-206?

Reviewer 3 Report

Summarized original contribution of the paper.

This manuscript provides UAV Swarm dataset that can be utilized on existing and future multiple object tracking algorithms. A manually annotated UAV swarm video data, entitled ‘UAVSwarm Dataset’, is provided where it consists of 13 different scenes and more than 19 types of UAV recorded, including 12598 images. The authors emphasize the importance of UAV object tracking algorithms for military defense purposes since UAV are now widely used in their combat missions working with ad-hoc networks which consist of hundreds of small-size and low-cost UAV. The authors imply that existing algorithms are not adequate for UAV swarms. The author’s approach is to provide a quality proven UAV swarm dataset that can be used to train multi-object tracking algorithms for future studies. Such approach may be useful when training multi-object tracking algorithms for UAV swarm in military-related applications. The author demonstrated two multi-object tracking algorithms, GNMOT and ByteTrack, to conduct comprehensive tests and performance verification on the dataset and evaluation metrics. While their comparison is well represented with tables and figures in general, the performance of object detection is somewhat missing; it is necessary to show the performance of UAV object detection in the validation to support the authors’ logic.

Specific and detailed comments:

As modern warfare is in rapid transition to an unmanned and intelligent system, UAVs are actively utilized in combat these days. Successful development and generation of the proposed work would contribute to substituting current radar-based surveillance systems in military applications. However, there are some deficiencies between the conducted research and the meaning of the results. In addition, this manuscript also has countless wording errors, grammatical errors, and unsupported sentences. Some sentences cannot be understood and are unfocused either because they are out of context, or there is poor English. This paper must be revised very carefully, but such revision is encouraged for future publication. The technical contribution would be nice if the details were brought forward in the text. Please see the detailed comments below.

  1. In Section 2. Related Works, the authors do not provide the number of UAVs in each provided dataset. Overall, the information given on the related dataset is not consistent in topics; it is confusing and hard to understand which parts to compare with. For example, for some datasets, the objects that are included in the dataset are mentioned where the annotation format problem is randomly aroused in one of the datasets.
  2. The authors mention the key issues of the current Multiple Object Tracking (MOT) and Multiple Target Tracking (MTT) as follows: frequent occlusion, Track initialization and termination, similar appearance, and interaction among multi-objectives. However, the authors do not provide how to deal with each of these key issues, as it is related to calibrating errors. The only part mentioned in the manuscript is frequent occlusion and track initialization.
  3. Poor organization is shown throughout the paper where sentences randomly appear in between the figures and tables and the figure names are not aligned with their corresponding figures, as well as inconsistent spacing.
  4. The performance of the dataset is validated using two multi-object tracking algorithms GNMOT and ByteTrack which are detection-based multi-object tracking. While the paper performs an experiment of a multi-object-tracking algorithm on the dataset and gives detailed results of the performances, it lacks any detail regarding the object-detection performances. For example, although it can be observed from the result figures that GNMOT gives high accuracy and reliable performance on tracking the multiple UAVSwarm Dataset, the dataset does not contain any other small flying objects such as birds, airplanes, that may create errors. It is difficult to determine whether this dataset also contains different aerial devices or flying objects that are highly related to the performance of object detection, not object-tracking.
  5. When introducing the ByteTrack algorithm, which uses detector YOLOX, the author states that it gives excellent performance when obtaining detecting results. However, when the ByteTrack results show error and low performance compared to running through MOTA, the authors say that the detection results generated by YOLOX are not very ideal for UAVs with small objects or fast-moving objects. The two comments on the performance of the algorithm contradict each other which gives doubts about the performance of the dataset.

Round 2

Reviewer 2 Report

The responses to the questions are satisfactory. I have no further questions.